# Proteomics Computational Analyses Suggest that the Antennavirus Glycoprotein Complex Includes a Class I Viral Fusion Protein (α-Penetrene) with an Internal Zinc-Binding Domain and a Stable Signal Peptide

**DOI:** 10.3390/v11080750

**Published:** 2019-08-14

**Authors:** Courtney E. Garry, Robert F. Garry

**Affiliations:** 1School of Nursing, Johns Hopkins University, Baltimore, MD 21205, USA; 2Department of Microbiology and Immunology, Tulane University School of Medicine, New Orleans, LA 70112, USA; 3Zalgen Labs, Germantown, MD 20876, USA

**Keywords:** arenavirus glycoproteins, antennavirus, mammarenavirus, hartmanivirus, reptarenavirus, class I viral fusion protein, internal zinc-binding domain, stable signal peptide

## Abstract

A metatranscriptomic study of RNA viruses in cold-blooded vertebrates identified two related viruses from frogfish (*Antennarius striatus*) that represent a new genus *Antennavirus* in the family *Arenaviridae* (Order: *Bunyavirales*). Computational analyses were used to identify features common to class I viral fusion proteins (VFPs) in antennavirus glycoproteins, including an N-terminal fusion peptide, two extended alpha-helices, an intrahelical loop, and a carboxyl terminal transmembrane domain. Like mammarenavirus and hartmanivirus glycoproteins, the antennavirus glycoproteins have an intracellular zinc-binding domain and a long virion-associated stable signal peptide (SSP). The glycoproteins of reptarenaviruses are also class I VFPs, but do not contain zinc-binding domains nor do they encode SSPs. Divergent evolution from a common progenitor potentially explains similarities of antennavirus, mammarenavirus, and hartmanivirus glycoproteins, with an ancient recombination event resulting in a divergent reptarenavirus glycoprotein.

## 1. Introduction

Until 2015, the *Arenaviridae* was comprised of a single genus of mammalian ambisense RNA viruses divided into two large monophyletic groups [1]. New World (NW) arenaviruses infect rodents of the family Cricetidae [2,3]. An exception is Tacaribe virus (TCRV) that infects *Artibeus* bats and may be vectored by insects [4]. NW arenaviruses from Northern America infect rodents from the subfamily Neotominae, whereas those from Latin American and the Caribbean infect rodents from the subfamiliy Sigmodontinae [5,6]. Old World (OW) arenaviruses are found in African rodents from the family Muridae subfamily Murinae [7,8,9,10]. Recently, OW arenaviruses have been isolated from mice and shrews in Asia [11,12,13]. Several arenaviruses spillover from their rodent hosts to humans where they can cause severe hemorrhagic fevers [14,15]. Lassa virus (LASV), an OW arenavirus, has been designated as a priority pathogen by the World Health Organization (WHO) [16] and the Coalition for Epidemic Preparedness and Innovations (CEPI) [17]. These rankings for LASV are based in part on the potential for further geographic expansion of its rodent reservoirs, the frequent importation to North America and Europe, and the possible emergence of novel strains in densely populated West Africa. 

In 2012, DeRisi and colleagues reported the isolation of a novel arenavirus and demonstrated that it was representative of viruses that can cause a disease in pet snakes worldwide [18]. Inclusion body disease (IBD) spreads among boid snakes (boas and pythons) in captivity [18,19,20,21,22,23,24]. Bacterial infections, neurological pathology, anorexia, and withering that result in death of the snakes are characteristics of IBD. Further metagenomic analyses identified numerous genetically diverse arenaviruses from captive snakes [21,25,26]. Two genera, *Reptarenavirus* and *Hartmanivirus*, were established in the *Arenaviridae* to include the snake arenaviruses [1,27,28]. While sometimes present in healthy snakes, reptarenaviruses are the causative agent of IBD [20,29,30], whereas hartmaniviruses have not been shown to cause disease [25]. With the discovery of these new arenaviruses, NW and OW arenaviruses were placed taxonomically into the genus *Mammarenavirus* [1]. Further expansion of the *Arenaviridae* was necessitated based on results of an extraordinary large-scale metatranscriptomic study of RNA viruses in cold-blooded vertebrates that identified two arenaviruses from frogfish (*Antennarius striatus*) among hundreds of novel viruses [31]. Based on sequence similarity and phylogenetic analyses, Wēnlǐng frogfish arenavirus-1 (WlFAV-1, striated antennavirus) and Wēnlǐng frogfish arenavirus-2 (WlFAV-2, hairy antennavirus) represent a new genus, *Antennavirus* [32].

Genetic differences are apparent when comparing the mammalian, reptilian, and fish arenaviruses. Mammarenaviruses, reptarenaviruses, and hartmaniviruses have bi-segmented genomes [33]. The small segment is ambisense and encodes the glycoprotein complex (GPC) and the nucleocapsid protein (NP). The large segment encodes the RNA-dependent RNA polymerase (L) and, in mammarenaviruses and reptarenaviruses in which the segment is ambisense, a zinc-binding or matrix protein (Z). Sequencing of the genome of the prototype hartmanivirus, Haartman Institute snake virus (HISV), showed that this virus does not encode a Z protein [26]. This unexpected result was confirmed in subsequent isolates of related hartmaniviruses that also lack a gene encoding Z [25]. Z participates in several processes in the virus replication cycle, including the formation of infectious viral particles and viral budding [34], indicating that hartmaniviruses utilize alternative mechanisms. In contrast, the frogfish arenavirus contains three genome segments [31].

The LASV GPC is a trimer of heterodimers, each containing receptor-binding glycoprotein 1 (GP1) and a transmembrane fusion protein GP2 [35]. LASV also encodes an unusual long stable signal peptide (SSP) that is required for proper processing of GPC and is retained in the virion as part of the complex. The GPC precursor is trafficked from the endoplasmic reticulum to the Golgi, where it is heavily N-glycosidated and processed by cellular proteases (SPase, SKI1/SP1) into its mature form comprising non-covalently linked GP1, GP2, and SSP. Reptarenavirus L and NP proteins share 17%–26% pairwise amino acid identity with the L and NP proteins of OW and NW mammarenaviruses [18,33]. However, the reptarenavirus glycoproteins did not contain similarity to other arenavirus glycoproteins detectable by commonly used search programs. Instead, reptarenavirus glycoproteins were related to the glycoproteins of filoviruses (e.g., Ebola and Marburg viruses), avian retroviruses (e.g., avian leukosis virus), and cellular syncytin, the product of a repurposed endogenous retroviral gene [18]. No similarities with arenavirus or filovirus glycoproteins or other proteins could be detected for the putative glycoprotein sequence of the frogfish arenaviruses, whereas antennavirus L and NP proteins show homology to L and NP proteins of other arenaviruses [31,33]. Here, we examine the amino acid sequences of proteins encoded by the newly described arenaviruses using computational tools that have previously proven potent in identifying structural features of viral fusion proteins [36,37,38,39,40,41]. 

## 2. Materials and Methods

### 2.1. Sequences

GenBank accession numbers of the arenavirus and filovirus proteins used for sequence and structural analyses are provided in Appendix A.

### 2.2. Proteomics Computational Methods

Pairwise alignments were performed using William Pearson’s Lalign program [42] that implements the algorithm of Huang and Miller [43]. To generate alignments between multiple sequences we used Clustal Omega, a part of the European Molecular Biology Laboratory-European Bioinformatics Institute (EMBL-EBI) search and sequence analysis tool kit that uses seeded guide trees and hidden Markov model (HMM) profile–profile techniques [44]. Secondary structure, solvent accessibility, transmembrane helices, globular regions, coiled-coil regions, and other protein features were analyzed using PredictProtein [45]. Domains with significant propensity to form transmembrane helices were identified with TMpred v. 1.0 [46]. TMpred is based on a statistical analysis of TMbase, a database of naturally occurring transmembrane glycoproteins [47]. Topology and other features of membrane proteins were explored by means of hydropathy plots with the Membrane Protein Explorer from the Stephen White laboratory [48]. The presence of signal peptides and the location of their cleavage sites was analyzed using SignalP (v5.0), which is based on deep convolutional and recurrent neural network architecture including a conditional random field [49]. O-linked glycosylation sites were predicted by Net-O-Glyc v. 4.0, which produces neural network predictions of mucin-type GalNAc O-glycosylation sites in mammalian proteins [50].

## 3. Results

### 3.1. Antennavirus Glycoprotein 2 Has Features of a Class I Viral Fusion Protein

Shi and colleagues [31] used metatranscriptomic sequencing to identify two frogfish arenaviruses, WlFAV-1 and WlFAV-2, among 214 novel vertebrate-associated viruses of reptiles, amphibians, lungfish, ray-finned fish, cartilaginous fish, and jawless fish. Unlike other arenaviruses, the antennaviruses have three genome segments. The middle segment of both viruses is ambisense and encodes a glycoprotein precursor and a shorter protein of unknown function. No apparent similarity of the glycoprotein precursor to other arenavirus glycoproteins was detected [33]. The uncleaved glycoprotein precursor of WlFAV-1 is 728 amino acids and that of WlFAV-2 is 635 amino acids. The proteins share 26.5% identical amino acids (55.9% chemically similar amino acids) in a 626 amino acid overlap (Figure 1a). Features found in other arenaviruses are present in both glycoproteins. Following furin-like cleavage sites are N-terminal fusion peptides that contain hydrophobic amino acids. The fusion peptide of WlFAV-1 contains a dicysteine as previously described for mammarenavirus (LASV and lymphocytic choriomeningitis virus, LCMV) GP2 [38]. The N-terminal helices (helical region 1, HR1) in GP2 of WlFAV-1 and WlFAV-2 and hartmanivirus GP2 contain numerous leucine and isoleucine residues. The C-terminal helices (HR2) of mammarenavirus, hartmanivirus, and antennavirus GP2 have an abundance of charged residues [38]. Like the mammarenavirus and hartmanivirus GPCs [25,51,52], antennavirus GPC includes an intracellular zinc-binding domain that follows the transmembrane (TM) domain. At least seven cysteine or histidine residues that can coordinate with zinc are present. Signal prediction algorithms predict WlFAV-1 and WlFAV-2 encode extended virion-associated stable signal peptides (SSP), both with C-terminal cysteine residues that likely coordinate with the zinc-binding domains [52]. We also constructed two-dimensional models of the SSP and GP2 of LASV (Figure 1b), HISV (Figure 1c), and WlFAV-2 (Figure 1d) to compare the structure of antennavirus GPC to other arenavirus glycoproteins. These models suggest that the overall structure of antennavirus GPC is similar to the GPCs of mammarenaviruses and hartmaniviruses. Similar sequences or common structural/functional motifs are located similarly.

### 3.2. Reptarenavirus Glycoprotein 2 Is a Filovirus-Like Viral Fusion Protein

The reptarenavirus GP2 appears to contain all structural features that are common to class I viral fusion proteins, including an N-terminal fusion peptide, two extended α-helices, an intrahelical loop, and a carboxyl terminal transmembrane domain (Figure 2). However, as noted previously [18], the reptarenavirus glycoprotein bears greater sequence similarity to the glycoproteins of filoviruses than to arenaviruses. Comparison of the Golden Gate virus (GGV) GP2 with Ebola virus (EBOV) GP2 shows a 27.9% similarity (55.3% chemically similar) in a 179 amino acid overlap (Figure 2a). However, sequence similarity alone is insufficient to establish whether the GGV GP2 more closely resembles the GP2 equivalents of filoviruses, retroviruses, or the cellular α-penetrene syncytin. Analysis of protein structural features identifies reptarenavirus GP2 as a filovirus-like VFP. The reptarenavirus glycoprotein contains a C-7X-C-C intrahelical loop (X = any amino acid), which is similar to the C-6X-C-C intrahelical loop of filoviruses. GP2 of GGV and EBOV each contain two N-linked glycans, both in the same relative locations (Figure 2b,c). Furthermore, both the reptarenavirus glycoprotein and the filovirus glycoprotein (Figure 2) [53] contain a domain with several aromatic amino acids prior to the transmembrane anchor. A membrane proximal aromatic domain is contained in class I fusion proteins of certain retroviruses (including HIV and other lentiviruses) [54] and coronaviruses [55] but has not previously been described in arenaviruses. 

Linear models of glycoprotein 2 from members of the four arenavirus genera and a filovirus demonstrate the structural similarities and differences (Figure 3). Common features include N-terminal fusion domains, two extended α-helices, an intrahelical loop, and a carboxyl terminal transmembrane domain. However, the intrahelical loops of mammarenavirus, hartmanivirus, and antennavirus glycoproteins contain 18 or more amino acids between a pair of cysteines, which is longer than the intrahelical loop of the reptarenavirus glycoprotein. The third cysteine in the intrahelical loop of reptarenavirus and filovirus glycoproteins, which links the GP1 and GP2 subunits of GPC, is not present in mammarenaviruses, hartmaniviruses, and antennavirus glycoproteins. Neither reptarenavirus nor filovirus glycoproteins have an intracellular zinc-binding domain or an extended virion-associated SSP, which are present in the GPCs of mammarenaviruses, hartmaniviruses, and antennaviruses.

### 3.3. Antennavirus Glycoprotein 1 Does Not Display Features Conserved in Glycoprotein 1 of Other Arenaviruses

GP1s of mammarenaviruses have a conserved structure, consisting of commonly placed α-helices and a β-sandwich that comprises the receptor-binding domain (Appendix A) [35,56,57]. There are two conserved cysteine linkages and two conserved N-linked glycans in both NW and OW arenavirus GP1. This scaffold accommodates binding to divergent receptors. OW arenaviruses and some NW arenaviruses use α-dystroglycan as a cellular receptor, while other NW arenaviruses use the transferrin receptor [58,59]. Reptarenavirus, hartmanivirus, or antennavirus GP1 do not appear to display any of the conserved features found in mammarenavirus GP1, nor do they have any discernable structural features in common with each other. Such features would manifest as a conserved pattern of dicysteine bonds or glycosylation sites (Figure 4). The distinct structures of the GP1 may be due to the requirement for binding to distinct fish or reptile cellular receptors.

EBOV GP1 contains a distinctive mucin-like domain with numerous O-linked glycans that may offer a shield against antibody recognition [60]. With the exception of Lujo virus (LUJV), no mammarenavirus GP1 is predicted to contain O-linked glycans (Figure 4). In contrast, GP1 of certain reptarenaviruses, such as University of Helsinki virus (UHV), contain several predicted O-linked glycosylation sites and numerous predicted N-linked glycosylation sites. Hartmanivirus and antennavirus GP1s also contain several O-linked glycosylation sites, but fewer predicted N-linked glycosylation sites. While computational models can predict potential glycosylation sites, confirmation by chemical analysis, crystallography, or other methods is required to establish that the sites contain glycans.

### 3.4. Antennavirus Nucleoprotein and Large Proteins Share Similarities to Hartmanivirus Nucleoprotein and Large Proteins

Previous analyses indicated that NP and L genes of WlFAV-1 and WlFAV -2 have similarities to NP and L genes of other arenaviruses and lead to the designation of a new genus *Antennavirus* within the *Arenaviridae*. Phylogenetic analyses suggested a closer relationship of antennaviruses to hartmaniviruses than to members of other arenavirus genera [33]. To further examine the relationship of the antennaviruses to other arenaviruses we performed alignments of the amino acid sequences of proteins from representative members of the four arenavirus genera. LASV NP has distinct N- and C-terminal domains connected by a flexible linker [61,62]. Hastie et al. [62] showed that the N-terminal domain binds viral RNA but not cellular mRNA and contains a deep basic groove that channels the single-stranded RNA. While numerous residues in the N-terminus of NP are highly conserved amongst all arenaviruses, antennaviruses and hartmaniviruses also share several similar sequences not present in NP of members from other genera (Figure 5). The C-terminus of the LASV NP functions as a double-stranded RNA-specific exonuclease [61]. Antiviral responses are initiated by dsRNA, and the NP endonuclease of NP could serve to evade the immune system. The C-terminal domain of antennavirus NP contains the active site of the endonuclease (DEDDh box) and a metal-binding domain both of which are conserved in the NP of arenaviruses of other genera (Appendix A). Antennavirus L proteins also show sequence similarities to other arenavirus L proteins (Appendix A). These analyses confirm the previously suggested phylogeny of the four arenavirus genera [33]. Like hartmaniviruses, the antennaviruses do not encode a zinc-binding protein (Z, matrix). Reptarenaviruses encode a Z protein with highly conserved residues for binding two metal ions (Appendix A). In contrast to Z of mammarenaviruses, Z of reptarenaviruses contains a hydrophobic transmembrane domain suggesting that it is membrane associated. 

## 4. Discussion

Based on common features, such as an amino-terminal fusion peptide, and computer algorithms that predict protein configurations, Gallaher and colleagues suggested [37] that the ectodomain of the HIV-1 transmembrane protein (TM, gp41) and TM proteins of other retroviruses, all fit the scaffold of the postfusion structure of influenza virus hemagglutinin 2 (HA2) [63]. This was the first evidence that enveloped viruses from different families employ a common mechanism for cell entry. While initially considered speculative, our retroviral TM model was confirmed later by X-ray crystallographic structural determinations [64,65,66]. Subsequently, it was observed that common structural motifs are present not only in myxovirus HA2 and retrovirus TM, but also in the GP2 proteins of filoviruses [36] and arenaviruses [38], the spike proteins (S) of coronaviruses [67,68,69] and paramyxoviruses F proteins [70], which are now referred to collectively as class I VFPs. The envelope glycoproteins of flaviviruses, alphaviruses, and bunyaviruses form the structurally distinct class II VFPs [71,72], whereas rhabdoviruses and herpesviruses, have class III VFPs [73,74,75]. Here, we present proteomic computational analyses suggesting that antennavirus GPC includes a class I VFP with an internal zinc-binding domain and a SSP.

The discovery of arenaviruses in fish establishes a long evolutionary history for this virus family [31]. Hartmaniviruses and antennaviruses appear more closely related to each other than to either mammarenaviruses or reptarenaviruses. Mammarenaviruses and reptarenaviruses may also share a common progenitor. The lack of Z protein in both antennaviruses and hartmaniviruses is consistent with this evolutionary scenario. The sequence similarities detected between GPCs of mammarenaviruses, hartmaniviruses, and antennaviruses are consistent with divergent evolution from a common class I VFP progenitor (Figure 6). In this scenario, acquisition of the SSP and intracellular zinc-binding domain in GPC followed divergence of members of the *Arenaviridae* from a virus encoding the common progenitor of arenavirus and filovirus glycoproteins and other class I VFPs. 

The entry proteins of the DNA containing baculoviruses, including group II nucleopolyhedroviruses (NPV) and granuloviruses, are class I VFPs [76]. However, baculoviruses in group I NPV are class III VFPs [40]. Orthomyxoviruses such as influenza virus encode class I VFPs, but members of the *Thogotovirus* genus of the *Orthomyxoviridae* encode class III VFPs. The examples of baculoviruses and orthomyxoviruses suggest that viruses within a family may diverge by acquiring VFPs of distinct classes through recombination events. Recombination has been shown to occur in reptarenaviruses [77]. While other evolutional pathways are possible, reptarenaviruses may have replaced the arenavirus GPC or a progenitor with a filovirus-like class I glycoprotein via recombination (Figure 6). An unlikely alternative scenario involves a highly convergent pathway involving a common arenavirus and filovirus glycoprotein progenitor evolving to a reptarenavirus glycoprotein bearing both extensive structural and sequence similarity to filovirus GP2 as well as loss of or failure to acquire the SSP and zinc-binding domain plus independent acquisition or failure to lose the third cysteine in the intrahelical loop.

Hypotheses that rodent-borne viruses, such as hantaviruses and arenaviruses, co-evolved with their hosts [78] are proving to be overly simplistic [79]. There is no credible genetic evidence for co-evolution of hantaviruses and their rodent reservoirs [80]. In Africa, genetic signatures of co-evolution of arenaviruses, if any existed, have been eliminated by multiple host-switching and lineage extinction events [81]. The current studies suggest that evolution of arenaviruses, particularly with regard to their glycoproteins, may not be a simple linear process. Characterization of arenaviruses from other fish or additional vertebrates may illuminate complex pathways for evolution/acquisition of arenavirus glycoproteins.

## 5. Conclusions

Divergent evolution from a common progenitor potentially explains similarities of antennavirus, mammarenavirus, and hartmanivirus glycoproteins, with an ancient recombination event resulting in a divergent reptarenavirus glycoprotein. Arenaviruses infecting snakes appear to have followed distinct evolutionary pathways. Analyses of sequence and structural similarities and differences suggest that mammarenaviruses and reptarenaviruses split from antennaviruses and hartmaniviruses and then acquired a zinc-binding protein. 

## Figures and Tables

**Figure 1 viruses-11-00750-f001:**
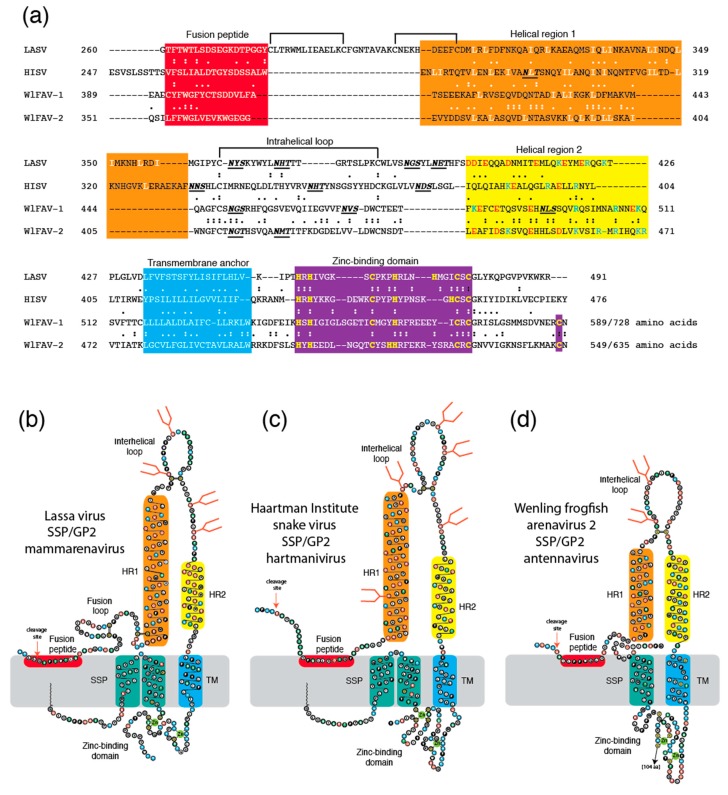
Comparisons of glycoproteins from representative members of three arenavirus genera. Panel (**a**): Amino acid sequence alignment of the glycoprotein 2 from Lassa virus (LASV, mammarenavirus), Haartman Institute snake virus (HISV, hartmanivirus), Wēnlǐng frogfish arenavirus-1 (WlFAV-1, striated antennavirus), and Wēnlǐng frogfish arenavirus-2 (WlFAV-2, hairy antennavirus). Panel (**b**): Two-dimensional model of LASV. Panel (**c**): Two-dimensional model of HISV. Panel (**d**): Two-dimensional model of HISV. Fusion peptide red, helical region 1 (HR1) orange, helical region 2 (HR2, yellow), aromatic domain (AD) green, transmembrane domain (TM) blue. Zinc-binding domain violet in Panel (**a**). Stable signal peptide (SSP) teal in Panels (**b**–**d**). Black lines: Dicysteine linkages. N-glycosylation sites: Bold italics in Panel (**a**) and orange stick figures in Panels (**b**–**d**). Similar amino acids are considered as hydrophobic: I,L,V,A,M,F; aromatic: W,F,Y; positive charged: K,R,H; negatively charged or chemically related: E,D,Q,N; special: G,P; polar: S,T.

**Figure 2 viruses-11-00750-f002:**
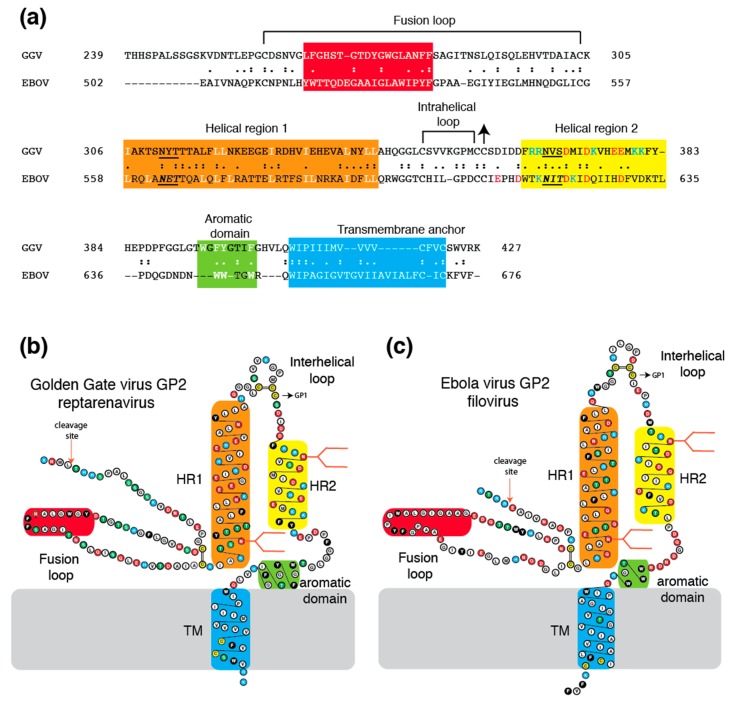
Comparison of glycoproteins of a reptarenavirus and a filovirus. Panel (**a**): Amino acid sequence alignment of glycoprotein 2 of Golden Gate virus (GGV, reptarenavirus) and Ebola virus (EBOV, filovirus). Panel (**b**): Two-dimensional model of GGV. Panel (**c**): Two-dimensional model of EBOV. Coloring and abbreviations as in Figure 1. Arrows indicate cysteine linkage to glycoprotein 1 (GP1).

**Figure 3 viruses-11-00750-f003:**
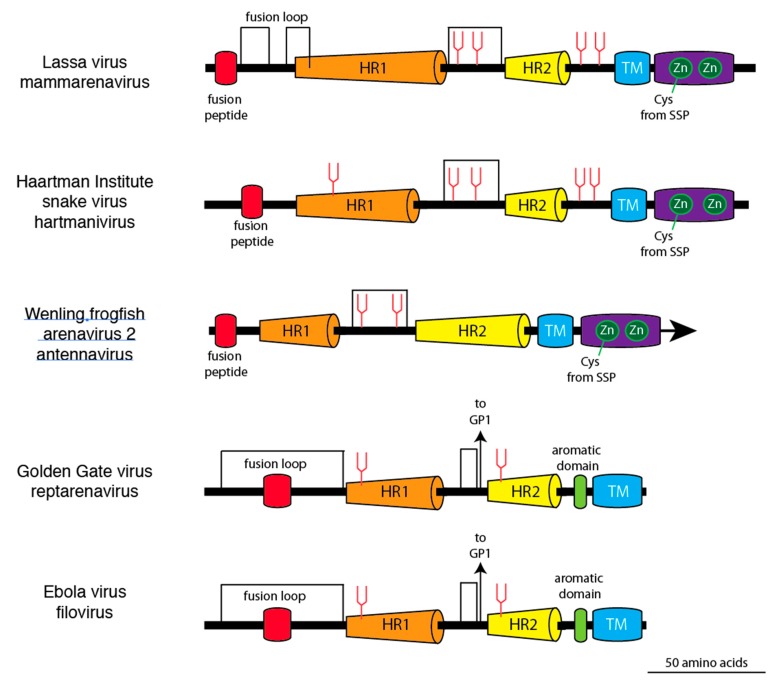
Linear models of glycoprotein 2 from representative members of the four arenavirus genera and a filovirus. Coloring and abbreviations as in Figure 1. Zn in green circle represents zinc-binding domains. A cysteine (Cys) from the stable signal peptide (SSP) coordinates with the first zinc-binding domain in Lassa virus, Haartman Institute snake virus, and Wēnlǐng frogfish arenavirus-2. Arrows indicate cysteine linkages to glycoprotein 1 (GP1) of Golden Gate virus and Ebola virus.

**Figure 4 viruses-11-00750-f004:**
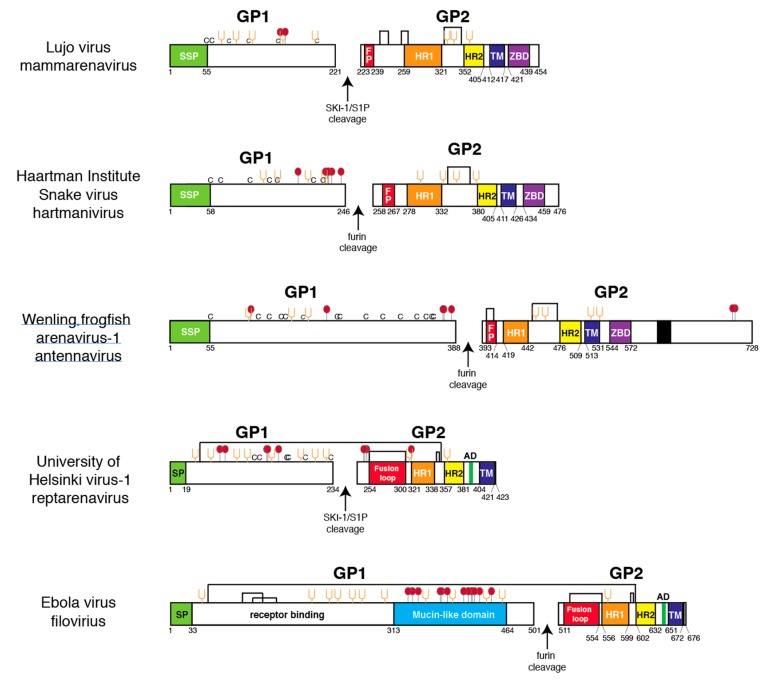
Linear models of glycoprotein 1 and 2 from selected members of the four arenavirus genera and a filovirus. Viruses were selected for the presence of potential O-linked glycosylation sites. Coloring and abbreviations as in Figure 1, except arrows represent cleavage sites by the indicated cellular proteases. Cysteine (C), fusion peptide (FP), zinc-binding domain (ZBD, aromatic domain (AD). Red balls: O-linked glycosylation sites. Amino acids are numbered from the N-terminus.

**Figure 5 viruses-11-00750-f005:**
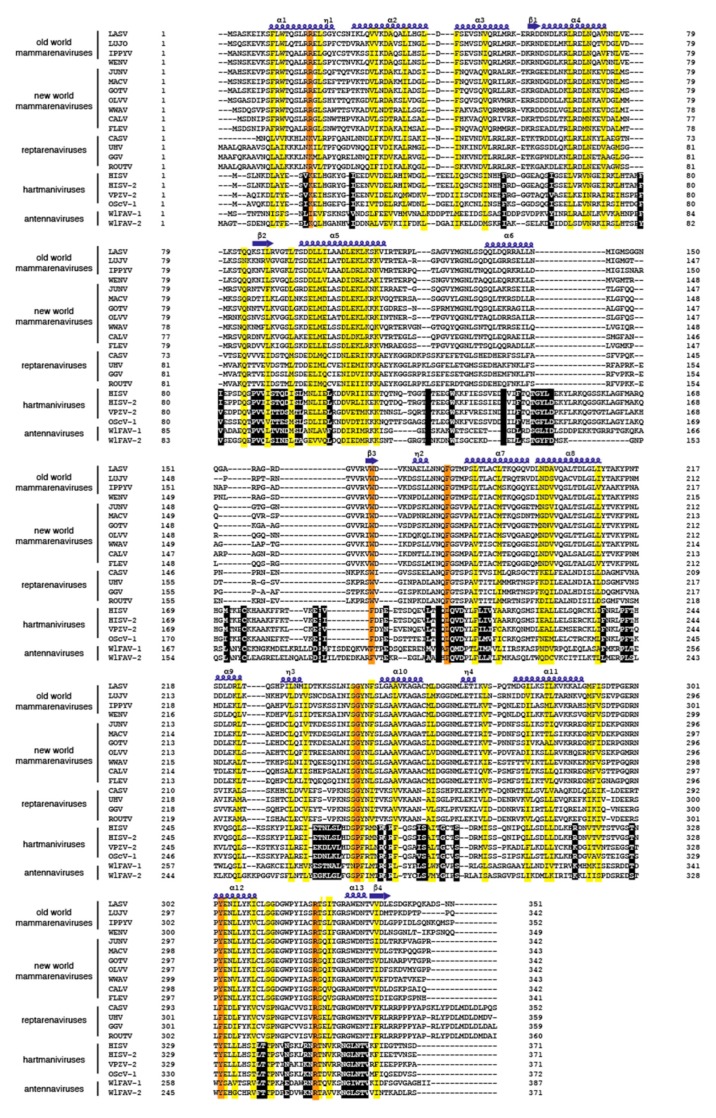
Amino acid sequence alignment of the N-termini of nucleoproteins from members of the four arenavirus genera. Virus abbreviations: Lassa, LASV; Lujo, LUJV; Ippy, IPPYV; Wenzhou, WENV; Junin, JUNV; Machupo, MACV; Guanarito, GOTV; Olivaros, OLLV; Whitewater Arroyo, WWAV, California (Pichinde), CALV; University of Helsinki, UHV; Golden Gate, GGV; Haartman Institute snake, HISV and HISV-2; Old Schoolhouse, OScV; Wēnlǐng Frogfish arenavirus, WlFAV-1 and WlFAV-2. Residues that are conserved (identical or chemically similar) are highlighted in yellow. Conserved residues involved in RNA binding are highlighted in orange. Blue springs represent helical structures and blue arrows are beta sheets. Similar residue/sequences in antennaviruses and hartmaniviruses not present in NP of members from other genera are highlighted in black.

**Figure 6 viruses-11-00750-f006:**
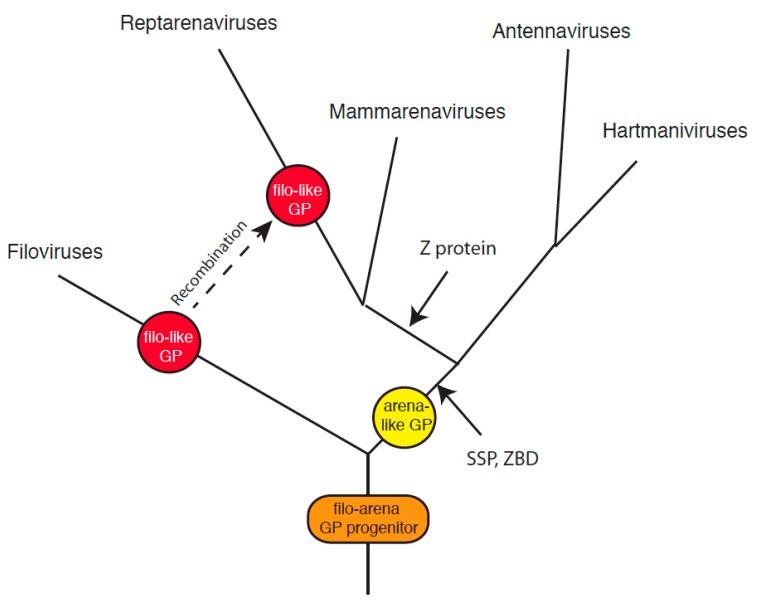
Possible evolution of the *Arenaviridae*. Members of the *Arenaviridae* and *Filoviridae* may have a common ancestor encoding the progenitor of arenavirus and filovirus glycoproteins. Acquisition of the stable signal peptide (SSP) and intracellular zinc-binding domain (ZBD) in the arenavirus glycoprotein complex (GPC) followed divergence from the filoviruses. Mammarenaviruses and reptarenaviruses acquired a zinc-binding protein (Z) following divergence from antennaviruses and hartmaniviruses. Reptarenaviruses then replaced the arenavirus GPC or a progenitor with a filovirus-like class I glycoprotein via recombination.

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
