# Peer review of "Proteomics Computational Analyses Suggest that the Antennavirus Glycoprotein Complex Includes a Class I Viral Fusion Protein (α-Penetrene) with an Internal Zinc-Binding Domain and a Stable Signal Peptide"

_viruses, 2019, doi:10.3390/v11080750_

Round 1

Reviewer 1 Report

In their manuscript Garry and Garry describe and extensive "in silico" analysis of the GPCs for family Arenaviridae viruses. I think the manuscript is written well (some minor things to revise) and the story is nice.

I have only minor comments, which are listed below:

line 45-46, references, please consider including: Hetzel U et al. J Virol. 2013 Oct;87(20):10918-35. and Hetzel et al. J Virol. 2014 Jan;88(2):1401.

line 51, hartmaniviruses have been isolated from or identified in snakes with BIBD, but it is unclear if hartmaniviruses are pathogenic

line 92, I believe CASV is just CAS virus, I think California Academy of Sciences did not want to have the virus called by their name. Why CASV and not Golden Gate virus (which is the type species for reptarenas)? Also why LASV and not LCMV?

Figure 2, please add explanations to the symbols presented in the figure. Most of the symbols are quite clear, but what does the arrow and GP1 imply? Please also spell out the abbreviations used (e.g. in the figure legend).

Figure 3, same as above. Also, if possibe, it would be nice to be able to see the amino acid letters (I realize this might be hard to execute), it would make the figure even nicer.

line 142-143, is the reference correct?

line 146, histidine *typo

Figure 4, same as for figures 2 and 3, i.e. the abbreviations.

Figure 5, please increase the size of the numbers. What are the orange stick figures? What are the "C"s above the proteins? Why were Lujo virus (instead of LASV) and UHV-1 (instead of CASV) selected for this figure?

Figure 6, should this appear after it has been mentioned in the text?

Line 276, I don't believe that recombination is actually that common, these snakes are practically always infected with multiple reptarenaviruses, yet the number of recombinant viruses (by phylogeny) is not that great. Recombination in negative-sense RNA viruses is less common than in positive-sense RNA viruses. I am also not a big fan of the hypothesized recombination between filo- and arenaviruses, however, as the authors point out the alternative scenario for reptarenavirus GPC origin seems unlikely too.

Author Response

We thank the reviewer for constructive comments. Our responses follow:

line 45-46, references, please consider including: Hetzel U et al. J Virol. 2013 Oct;87(20):10918-35. and Hetzel et al. J Virol. 2014 Jan;88(2):1401.

Apologies for excluding pertinent refernences - now included.

line 51, hartmaniviruses have been isolated from or identified in snakes with BIBD, but it is unclear if hartmaniviruses are pathogenic

line 92, I believe CASV is just CAS virus, I think California Academy of Sciences did not want to have the virus called by their name. Why CASV and not Golden Gate virus (which is the type species for reptarenas)? Also why LASV and not LCMV? 

We now use GGV,instead of CAS.

Figure 2, please add explanations to the symbols presented in the figure. Most of the symbols are quite clear, but what does the arrow and GP1 imply? Please also spell out the abbreviations used (e.g. in the figure legend).

Explanations to the symbols have been added.

Figure 3, same as above. Also, if possibe, it would be nice to be able to see the amino acid letters (I realize this might be hard to execute), it would make the figure even nicer. 

Explanations to the symbols have been added. Amino acids letter have been added.

line 142-143, is the reference correct?

No - but it has been corrected.

line 146, histidine *typo

Typo was fixed.

Figure 4, same as for figures 2 and 3, i.e. the abbreviations.

Abbreviations have been added to all figure legends.

Figure 5, please increase the size of the numbers. What are the orange stick figures? What are the "C"s above the proteins? Why were Lujo virus (instead of LASV) and UHV-1 (instead of CASV) selected for this figure? 

Size of the numbers has been increased (it looks better now). Organce stick figures explained. Lujo was used because it is the only mammarenavirus with predicted O-glycans.

Figure 6, should this appear after it has been mentioned in the text?

Order of figures is changed.

Line 276, I don't believe that recombination is actually that common, these snakes are practically always infected with multiple reptarenaviruses, yet the number of recombinant viruses (by phylogeny) is not that great. Recombination in negative-sense RNA viruses is less common than in positive-sense RNA viruses.

Acknowledged - we modified the text to reflect the reviewer's point.

I am also not a big fan of the hypothesized recombination between filo- and arenaviruses, however, as the authors point out the alternative scenario for reptarenavirus GPC origin seems unlikely too. 

Acknowledged - we modified the text to reflect alternative evolutionary scenarios.

Reviewer 2 Report

The manuscript by Garry and Garry describes computational analysis of multiple viruses in the Arenaviridae.  These findings include discovery and comparison of putative fusion sequences, zinc binding domains, and stable signal peptides.  Overall, the data are interesting but different viruses are compared in individual figures, making interpretation difficult.

Major concerns:

1)  Different viruses are compared in different figures, which makes consistent comparison difficult.  For example, Figure 1 compares LASV, HISV, WIFAV-1, and WIFAV-2, but Figure 2 has LASV, HISV, WIFAV-2, CASV, and EBOV, Figure 3 has LASV, HISV, and WIFAV-2, Figure 5 has Lujo, HISV, WIFAV-1, Helsinki-1, and EBOV, and Figure 6 has 21 different viruses.  A more uniform comparison in each Figure (when relevant) would make the presentation of data clearer.

2)  Some figures are hard to interpret:

Figure 1:  difficult to see bolded N-glycan sites

Figure 6:  difficult to see gray highlights or read individual sequences

3) Figure 1b should probably be moved to Figure 4, since both deal with comparisons with filoviruses

4)  Please discuss the difficulties of predicting of O-glycans (Section 3.3) in the results or discussion

5)  The conclusion section should be expanded to discuss the sequence analysis in more detail

Author Response

We appreciate the helpful comments of this reviewer.

Our responses follow:

1)  Different viruses are compared in different figures, which makes consistent comparison difficult.  For example, Figure 1 compares LASV, HISV, WIFAV-1, and WIFAV-2, but Figure 2 has LASV, HISV, WIFAV-2, CASV, and EBOV, Figure 3 has LASV, HISV, and WIFAV-2, Figure 5 has Lujo, HISV, WIFAV-1, Helsinki-1, and EBOV, and Figure 6 has 21 different viruses.  A more uniform comparison in each Figure (when relevant) would make the presentation of data clearer.

We changed the viruses represented in the figures to be more uniform. However, in Figure 5 (now Figure 4) we used viruses that were selected for their distinct glycosylation patterns. For example, LUJV is the only mammarenavirus with predicted O-glycans. This is now indicated more directly. Figure 6 (now Figure 5) does indeed have 21 different viruses, but this number of viruses is necessary to demonstrate the conservation of sequence across the genera.

2)  Some figures are hard to interpret:

Figure 1:  difficult to see bolded N-glycan sites

We added  underlines that highlight the N-glycan sites better.

Figure 6:  difficult to see gray highlights or read individual sequences. Now Figure 5 - Grey highlights were changed to black. Figure is printed at highest resolution possible.

3) Figure 1b should probably be moved to Figure 4, since both deal with comparisons with filoviruses.

We moved 1b to former figure 4 (now Figure 3). We moved 1a to former Figure 2 which is now Figure 1. We thank the reviewer for suggesting these changes, which clarifies the presentation of the data.

4)  Please discuss the difficulties of predicting of O-glycans (Section 3.3) in the results or discussion

A disclaimer was added about predicting O-glycans.

5)  The conclusion section should be expanded to discuss the sequence analysis in more detail

We added a brief discussion of this point to the conclusion.

Round 2

Reviewer 2 Report

Revisions have satisfied the concerns of this reviewer.